# InvariantCloud: A Globally Invariant, Uniquely Indexed Point Cloud Framework for Robust 6-DoF Tactile Pose Tracking

Pengfei Ye[1,†], Yuxiang Ma[2,†], Yi Zhou[1], Molong Duan[1]

[1]Department of Mechanical and Aerospace Engineering, The Hong Kong University of Science and Technology
[2]Department of Mechanical Engineering, Massachusetts Institute of Technology

*Abstract*—Recent advances in imitation learning and vision–language models highlight the need for high-fidelity tactile perception, with 6-DoF tactile object pose estimation providing a crucial foundation for precise robotic manipulation. We introduce InvariantCloud, a 6-DoF pose estimation framework that leverages the global invariance of surface marker constellations on vision-based tactile sensors. In contrast to recent approaches, our one-shot globally invariant point cloud registration suppresses cumulative drift and overcomes long-standing limitations in accurately estimating yaw (Z-axis) rotation. Experimental verifications show that InvariantCloud achieves sub-2° yaw tracking error and sub-1.5° yaw re-localization repeatability, demonstrating its superior precision and robustness.

*Index Terms*—Globally Invariant Point, 6-DoF Pose Estimation, Tactile Sensing

## I. INTRODUCTION

Vision-based tactile sensors are essential for high-precision manipulation [1], [2], yet accurate 6-DoF pose tracking remains challenging due to cumulative drift and weak Z-axis rotation observability. Existing methods rely on frame-to-frame registration (Lucas-Kanade optical flow, ICP) or normal-field optimization [3], both prone to drift accumulation and failure on locally planar or quasi-isotropic surfaces where yaw becomes unobservable.

We present InvariantCloud (Fig. 1), a globally invariant point cloud framework that assigns unique IDs to dense surface markers. This enables direct correspondence matching across frames without nearest-neighbor search, eliminating registration ambiguity. Combined with Kabsch SVD for XY rotation and PCA principal-axis extraction for Z rotation, our method achieves sub-2° yaw tracking error, sub-1.5° repeatability, and robust long-horizon tracking. Our contributions are: (1) a globally invariant, uniquely indexed point cloud for drift-free 6-DoF pose estimation; (2) a PCA-based Z-axis solver that exploits global marker layout stability for reliable yaw recovery.

## II. METHOD

To enable precise tracking of object poses without requiring prior 3D models, a globally invariant point cloud is leveraged on the surface of a vision-based tactile sensor. A flat, dense reference point cloud is first acquired under a no-contact condition, with a unique identifier (ID) assigned to each point. This spatially stable and uniquely indexable global reference allows direct establishment of inter-frame correspondences

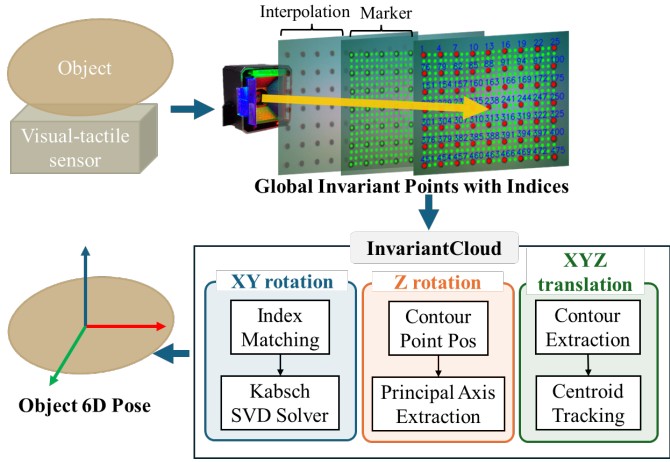

Fig. 1: A schematic overview of the proposed 6D pose tracking framework, illustrating global invariant point generation, index matching, Kabsch SVD solving for XY rotation, principal axis extraction for Z-axis rotation, and centroid tracking for XYZ translation.

via point IDs, effectively eliminating errors in point cloud registration.

### A. Dense Reference Cloud Construction and XY Rotation

A globally referenced point cloud with unique identifiers is constructed by first reconstructing a high-accuracy height map from sphere calibration contacts. An MLP predicts surface normal tilt angles from RGB values and pixel coordinates, incorporating information from $N_{markers}$ intrinsic fiducial markers [1], [4]. We establish bidirectional pixel-to-3D mapping [5] where each pixel location is uniquely mapped into the global 3D coordinate system. The $N_{markers}$ fiducial markers arranged in an $R_{orig} \times C_{orig}$ grid are densified via bilinear interpolation to generate a high-density point cloud (typically several hundred points), with each interpolated point receiving a globally unique identifier.

For each frame, we apply dual thresholds on color difference and height depression, followed by morphological operations to produce a stable contact mask $C$. Since each 3D point carries a unique global identifier, one-to-one correspondences are established directly via IDs. The paired point sets are passed to the Kabsch algorithm [6] for closed-form SVD solution. After centering both point clouds, the optimal rotation $R^\star \in SO(3)$ is obtained via:

$$H = \sum_{i=1}^{N}(p_i - \bar{p})(q_i - \bar{q})^\top \qquad (1)$$

$$R^\star = V\,\mathrm{diag}(1,1,\mathrm{sign}(\det(VU^\top)))U^\top \qquad (2)$$

where $H = U\Sigma V^\top$ is the SVD decomposition.

### B. PCA Principal Axis Solver for Z Rotation

To address Z-axis rotation estimation difficulty, an innovative PCA principal axis solver is proposed. The method exploits: (i) high stability of invariant points under repeated contacts, and (ii) sensitivity of contact silhouette to yaw. Except for ideal spheres, yaw rotation alters the contact outline and thus the subset of enclosed global points. At each frame, invariant points within the current silhouette are extracted and their principal axis is computed via PCA [7], [8]. The contact subset $\mathcal{S}_t = \{p_i \in \mathcal{P} \mid C_t(p_i) = 1\}$ is extracted, covariance matrix $M_t$ is computed via eigen decomposition, and the principal axis $a_t$ yields Z-axis rotation angle $\theta_t = \mathrm{atan2}(a_{t,y}, a_{t,x})$.

### C. Contact Centroid-Based Translation

To track 3D translation, we use the geometric centroid of the current contact subset [9]. We apply morphological closing to obtain a stable dominant contour, compute its area-moment centroid, and lift it to 3D. Frame-to-frame centroid differences yield XY translation, while the Z component is estimated from contact region height.

## III. EXPERIMENTS AND RESULTS

This section evaluates the performance of InvariantCloud for 6-DoF pose tracking on common household objects (Fig. 1). All experiments used the GelSight Mini visual-tactile sensor [1] (resolution: 320×240, frame rate: ≈25 Hz), with the reference point cloud upsampled to a 19×25 grid (475 points). Two baselines are compared: (1) Lucas-Kanade optical flow [10] with nearest-neighbor ICP [11], [12], and (2) NormalFlow [3] using Gauss-Newton optimization on surface normals.

We define three metrics: (1) *Static cumulative drift*: accumulated pose deviation while the object remains stationary; (2) *Repeatability error*: deviation when returning to initial pose after rotation/translation; (3) *Tracking accuracy*: deviation between estimated and ground-truth motion across a known range.

### A. Static Cumulative Drift and Repeatability

Table I shows cumulative error over one minute of stationary contact. Both NormalFlow and InvariantCloud exhibit excellent static performance, while ICP gradually accumulates drift. For repeatability, objects were rotated or translated along single axes and returned to near-initial poses. Five trials per object were averaged. ICP shows significant errors even in static conditions. NormalFlow maintains reasonable X/Y-axis accuracy but exhibits up to 20° error in Z-axis rotation for elliptical contacts (e.g., eggs). InvariantCloud consistently achieves repeatability errors below 1.5° across all objects.

### B. Z-Axis Rotation Tracking Accuracy

ICP fails for Z-axis tracking due to surface slip during rotation. We compare NormalFlow and InvariantCloud for 90° counterclockwise Z-axis rotation across four objects, with

TABLE I: Static Cumulative MAE During One-Minute Stationary Contact

| Method | x(mm) | y(mm) | z(mm) | $\theta_x(^\circ)$ | $\theta_y(^\circ)$ | $\theta_z(^\circ)$ |
|---|---|---|---|---|---|---|
| **InvariantCloud** | **0.22** | **0.14** | **0.11** | **0.62** | **0.69** | **0.84** |
| NormalFlow | 0.41 | 0.38 | 0.28 | 1.42 | 0.97 | 0.78 |
| ICP | 2.52 | 3.14 | 2.29 | 10.89 | 8.15 | 7.19 |

motion capture providing ground truth (Fig. 2). NormalFlow performs well on objects with distinct contours (scissors, sensor box) but fails on less distinctive surfaces (egg, utility knife elliptical end), where estimated angles remain nearly static. InvariantCloud achieves robust tracking across all objects, with tracking error consistently below 2°, even for challenging quasi-isotropic surfaces [13], [14]. This demonstrates the effectiveness of our PCA principal-axis method in exploiting global marker layout stability for reliable object pose estimation [15].

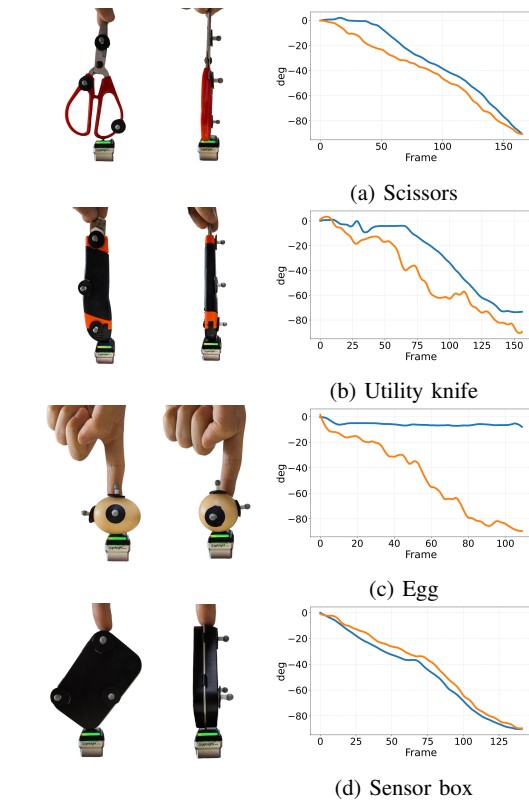

(a) Scissors

(b) Utility knife

(c) Egg

(d) Sensor box

Fig. 2: Z-axis rotation tracking comparison. Blue: NormalFlow; Yellow: InvariantCloud. Objects rotated counterclockwise from 0° to -90°.

## IV. CONCLUSION

We present InvariantCloud, a 6D pose tracking framework based on globally invariant point clouds with unique IDs. By enabling direct inter-frame correspondence via ID matching and combining Kabsch SVD with PCA principal-axis extraction, our method achieves sub-2° yaw tracking error and sub-1.5° repeatability, outperforming existing baselines. A limitation remains for perfectly spherical objects where Z-axis rotation is unobservable.

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
