# OpenReview forum: "InvariantCloud: Globally Invariant Point Cloud  Registration for High-Precision  6DoF Tactile Pose Tracking"
_IEEE.org/IROS/2025/Workshop/Tactile_Sensing — IROS 2025 Workshop Tactile Sensing OralPoster_

### Official Review · Reviewer_YFE5 · 2025-09-16
**A very robust tactile pose tracking work**

**Rating:** 9
**Confidence:** 4

**Review:**

The authors presented a novel object pose tracking method based on vision-based tactile sensing, and demonstrated the high tracking precision and robustness through comprehensive evaluation. A very solid work!

---

### Official Review · Reviewer_ZKr7 · 2025-09-23
**A solid and drift-robust tactile pose tracking method**

**Rating:** 9
**Confidence:** 4

**Review:**

This paper presents a novel method for 6-DoF tactile pose tracking using a globally invariant point cloud with unique IDs, effectively addressing the critical problem of cumulative drift in long-horizon tracking, particularly for the challenging Z-axis rotation. The work is technically sound,  with clear metrics and convincing results that demonstrate superior performance.
﻿
A minor recommendation for improvement would be to include a brief runtime analysis or a mention of the computational efficiency of the proposed ID-based matching compared to traditional ICP or optical flow, as this is highly relevant for real-time robotic applications.

---

### Official Review · Reviewer_E35v · 2025-09-23
**Review of submission #23**

**Rating:** 9
**Confidence:** 4

**Review:**

The paper proposes a novel method for 6DOF pose tracking with vision-based tactile sensor (VBTS), specifically GelSight mini.

Strengths:
* The method InvariantCloud is well-explained and thoroughly tested.
* The experiments evaluate the method on a variety of objects on different aspects, such as repeatability, accuracy, etc.
* The paper is well-written and easy to follow.

Weakness/Questions:
* The paper focuses on tracking with GelSight mini sensor. A brief discussion on the possibility of generalizing the method to sensors of different resolutions, different sensor types would be helpful.